# Improving Adversarial Transferability via Intermediate-level Perturbation Decay

**Qizhang Li[1,2], Yiwen Guo[3]\*, Wangmeng Zuo[1]\*, Hao Chen[4]**
[1]Harbin Institute of Technology, [2]Tencent Security Big Data Lab, [3]Independent Researcher, [4]UC Davis
`{liqizhang95,guoyiwen89}@gmail.com`  `wmzuo@hit.edu.cn`  `chen@ucdavis.edu`

## Abstract

Intermediate-level attacks that attempt to perturb feature representations following an adversarial direction drastically have shown favorable performance in crafting transferable adversarial examples. Existing methods in this category are normally formulated with two separate stages, where a directional guide is required to be determined at first and the scalar projection of the intermediate-level perturbation onto the directional guide is enlarged thereafter. The obtained perturbation deviates from the guide inevitably in the feature space, and it is revealed in this paper that such a deviation may lead to sub-optimal attack. To address this issue, we develop a novel intermediate-level method that crafts adversarial examples within a single stage of optimization. In particular, the proposed method, named *intermediate-level perturbation decay (ILPD)*, encourages the intermediate-level perturbation to be in an effective adversarial direction and to possess a great magnitude simultaneously. In-depth discussion verifies the effectiveness of our method. Experimental results show that it outperforms state-of-the-arts by large margins in attacking various victim models on ImageNet (**+10.07%** on average) and CIFAR-10 (**+3.88%** on average). Our code is at https://github.com/qizhangli/ILPD-attack.

## 1 Introduction

Adversarial examples can be crafted without getting access to essential information (*e.g.*, architecture and parameters) of the victim models [34]. Due to their applicability of assessing AI safety in many real-world deployment environments, recent years have witnessed a steady increase in the study of black-box adversarial examples [29, 4, 11, 46, 8, 16]. A considerable proportion of black-box attacks rely on transferability of adversarial examples, *i.e.*, a phenomenon where adversarial examples crafted targeting a white-box substitute model can also fool unknown victims with a moderate success rate.

Despite being crucial to many attacks, the adversarial transferability has not been fathomed out. Some work attempts to regulate the intermediate-level perturbation (discrepancy between the intermediate-level representation of adversarial examples and that of benign examples) to show a large magnitude and align with an directional guided that can effectively maximize the prediction loss of a model [20, 26, 43, 52]. To achieve this, they choose to maximize the scalar projection of the intermediate-level perturbation onto a directional guide. The determination of the directional guide of such methods are various, and each was well-motivated. Though effective, such an objective inevitably deviates the intermediate-level perturbation from the well-motivated directional guide (as will be discussed in Section 3.1), and it may deteriorate the attack performance on the substitute model and lead to sub-optimal transferability despite showing a large perturbation magnitude in the middle layer.

In this paper, we propose a method that encourages the intermediate-level perturbation to possess a greater magnitude than a directional guide by its nature and to be in the same adversarial direction as that of the guide. This is achieved by introducing *intermediate-level perturbation decay (ILPD)*

---

\*Yiwen Guo and Wangmeng Zuo lead the project. Correspondence to: Wangmeng Zuo (wmzuo@hit.edu.cn).

in a single stage of optimization. Extensive experiments were conducted to show that our ILPD outperforms existing state-of-the-arts by large margins in attacking various victim models on CIFAR-10 and ImageNet. Moreover, such a method can further be readily integrated into a variety of existing transfer-based attacks.

## 2 Background and Related Work

Adversarial examples can be generated in white-box and black-box settings.

**White-box attacks.** In white-box settings, the architecture and parameters of the victim model are known to the adversary, and the gradient of the victim models can therefore be directly utilized to generate adversarial examples. Given a victim model $f : \mathbb{R}^n \to \mathbb{R}^c$ which has been trained to classify any input, a typical way of generating adversarial examples is to maximize the prediction loss $L(f(\mathbf{x} + \mathbf{\Delta x}), y)$ of $f$ while constraining the $\ell_p$ norm of the perturbation $\mathbf{\Delta x}$ to a benign input $\mathbf{x}$, *i.e.*, $\max_{\|\mathbf{\Delta x}\|_p \leq \epsilon} L(f(\mathbf{x} + \mathbf{\Delta x}), y)$, where $\epsilon$ is the perturbation budget and $y$ is the ground-truth. FGSM [14] derives a single gradient step to obtain the perturbation $\mathbf{\Delta x} = \epsilon \cdot \text{sign}(\nabla_{\mathbf{x}} L(f(\mathbf{x}), y))$ in an $\ell_\infty$ setting. If not otherwise mentioned, we denote $L(f(\cdot), \cdot)$ as $L_f(\cdot, \cdot)$ for simplicity of notation. For performing stronger attacks, iterative methods have also been proposed, *e.g.*, I-FGSM [25] and PGD [32] which update the perturbation at the $j$-th iteration as

$$\mathbf{\Delta x}_j = \text{Clip}^\epsilon_{\mathbf{\Delta x}}(\mathbf{\Delta x}_{j-1} + \eta \cdot \text{sign}(\nabla_{\mathbf{x}} L_f(\mathbf{x} + \mathbf{\Delta x}_{j-1}, y))), \tag{1}$$

where $\eta$ is the update step size and the $\text{Clip}^\epsilon_{\mathbf{\Delta x}}$ operation ensures that the norm of perturbation does not exceed $\epsilon$.

**Black-box attacks.** Issuing attacks in black-box settings is much more challenging than in white-box settings due to the lack of information of the victim model. In general, neither the architecture nor the parameter of the victim model is accessible in the black-box settings, and the attacker can only obtain the prediction confidence or even only the predicted label, providing any input. Two lines of methods have been developed for crafting adversarial examples under such circumstances, *i.e.*, query-based methods and transfer-based methods. Query-based methods advocate searching along promising directions [4, 5, 49, 48] or estimating the gradient of victim models via zeroth-order optimization [6, 22, 23, 41], while transfer-based methods train substitute models to craft adversarial examples and can be query-free.

**Adversarial transferability.** As the core of transfer-based attacks, the transferability of adversarial examples from substitute models to unknown victim models has been studied widely over recent years, and a variety of methods have been proposed to enhance it. Related to our work, some previous work also achieves improved transferability by modifying the gradient computation on substitute models. For instance, Guo *et al.* [15] advocated removing non-linear activations during back-propagation, especially in later layers of DNNs. Wu *et al.* [45] suggested to take gradients more through skip connections, if exist, on the substitute models. In particular, intermediate-level representations were taken special care of in some prior arts [20, 26, 16, 43, 52].

## 3 Perturbation Decay in the Feature Space

We revisit some existing attacks that operate on intermediate-level representations and identify their possible limitations in Section 3.1 and propose a novel method to improve them in Section 3.2.

### 3.1 Revisiting Intermediate-level Attacks

As transfer-based attacks, intermediate-level attacks are performed with at least a substitute model. Given a model input $\mathbf{x}$, the output of an $\alpha$-layer substitute model can be simplified as $f(\mathbf{x}) = W_\alpha^T \phi_{\alpha-1}(W_{\alpha-1}^T \ldots \phi_1(W_1^T \mathbf{x}))$, where, for $i \in \{1, \ldots, \alpha\}$, $\phi_i$ is an activation function, and $W_i \in \mathbb{R}^{n_{i-1} \times n_i}$ parameterizes a layer in the model. Assume that the input is of $n$ dimensions and there are $c$ classes for prediction, then we have $n_0 = n$, and $n_\alpha = c$. Existing intermediate-level attacks decompose the model into a $\beta$-layer feature extractor $h$ and an $(\alpha - \beta)$-layer classifier $g$, *i.e.*, we can formulate the model as $f(\mathbf{x}) = g(h(\mathbf{x}))$ for any $\mathbf{x}$. A motivation of intermediate-level attacks is that different models might share the same low-level representations in $h$ and it suffices to operate on the output layer of $h$ [20, 26].

A typical work of intermediate-level attack is ILA [20]. It attempts to regulate the intermediate-level perturbation (*i.e.*, discrepancy between the intermediate-level representation of adversarial examples and that of benign examples) to show a large norm and to be close to a directional guide. The directional guide in ILA is chosen as the discrepancy from benign representations that could be reached in the output space of $h$ and leads to maximum prediction loss of $f$, given the perturbation budget. Since white-box attacks all devote to achieving maximal prediction loss, ILA can be performed with simply a $u$-step baseline attack (*e.g.*, I-FGSM) and the intermediate-level perturbation achieved by the baseline attack as the directional guide.

Let $\mathbf{x} + \mathbf{\Delta x}_j$ be the adversarial input achieved at the $j$-th step of the baseline attack, we denote $\mathbf{z}_h^{(j)} := h(\mathbf{x} + \mathbf{\Delta x}_j)$ as the intermediate-level representation of $\mathbf{x} + \mathbf{\Delta x}_j$. Further, denote by $\mathbf{v}$ the directional guide, then we can formulate it as $\mathbf{v} = (\mathbf{z}_h^{(u)} - \mathbf{z}_h^{(0)})$.

ILA maximizes the scalar projection, which is

$$\max_{\|\mathbf{\Delta x}\|_p \leq \epsilon} \mathbf{v}^T(h(\mathbf{x} + \mathbf{\Delta x}) - h(\mathbf{x})), \tag{2}$$

where $1/\|\mathbf{v}\|_2$ is omitted as it does not affect the optimal solution. With such a learning objective, the norm of the vector projection is maximized if $\mathbf{v}$ and $h(\mathbf{x} + \mathbf{\Delta x}) - h(\mathbf{x})$ form an acute angle. Since the direction of intermediate-level perturbation is not restricted to be in exactly the same direction as that of the directional guide, the obtained intermediate-level perturbation often deviates from the directional guide in practice. Yet, the adversarial perturbations are generally not robust enough [1], and such directional deviations normally leads to decrease in the prediction loss of the substitute model, even with large perturbation magnitude in the middle layer.

A quick experiment was conducted to examine the impact of such directional deviations. The experiment was conducted on CIFAR-10 [24] using a VGG-19 [36] as the substitute model and another VGG-19 with the same $h$ as the victim model, just to simplify the analyses. The directional guide $\mathbf{v}$ of ILA was obtained by 10-steps I-FGSM following prior work [26, 16, 27]. We evaluated the cross-entropy prediction loss of the models with various intermediate-level perturbations to demonstrate how the attack performance can be affected by the scalar projection (onto $\mathbf{v}$) and the directional deviation (from $\mathbf{v}$). Somewhat unsurprisingly, as depicted in Figure 1, when fixing the direction of intermediate-level perturbations, the perturbations with larger magnitude leads to more significant attack per-

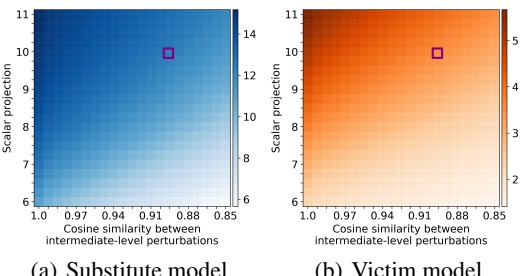

(a) Substitute model     (b) Victim model

Figure 1: How the prediction loss of (a) the substitute model and (b) the victim model vary with the scalar projection onto $\mathbf{v}$ and the direction deviation from $\mathbf{v}$ (*i.e.*, the cosine similarity between $\mathbf{v}$ and $h(\mathbf{x} + \mathbf{\Delta x}) - h(\mathbf{x})$). The purple box represents the results achieved by ILA.

formance on both the substitute model and the victim model. More enticingly, with intermediate-level perturbations showing similar scalar projection, those deviating more from the directional guide leads to lower prediction loss on the victim model and hence worse black-box attack performance. As illustrated in the figure, the ILA perturbation only slightly deviates from the directional guide $\mathbf{v}$ (with a cosine similarity of roughly 0.90), yet its attack performance is already inferior to that of the intermediate-level perturbations along the direction of $\mathbf{v}$ **with even smaller magnitude**. Follow-up work of ILA, *e.g.*, [26, 16], mostly focuses on developing more powerful directional guides, yet the deviation remains and the same problem persists.

## 3.2   Our Solution

Given Figure 1, one may consider that a straightforward improvement of the intermediate-level attacks is to instead maximize the magnitude of (intermediate-level) perturbation only along the direction of $\mathbf{v}$. Nevertheless, most existing intermediate-level attacks consists of two separate stages which sequentially seeks $\mathbf{v}$ and maximizes perturbation magnitude, and they fail to take into account whether there exists a larger intermediate-level perturbation along the direction of $\mathbf{v}$, given the perturbation budget in the input space of $f$.

In this context, we aim to develop a novel method that seeks intermediate-level perturbations to be in an adversarial direction and to possess larger norms (or say larger magnitudes) **compared to a directional guide in the same direction** simultaneously. As with previous methods, we consider a direction to be adversarial if a guide along this direction achieves large prediction loss by itself when being fed to the substitute classifier $g$. The intermediate-level perturbation with a relatively larger magnitude can simply be obtained by amplifying the directional guide by $\gamma\times$ in the middle layer. However, such an intermediate-level perturbation may not be reachable with a limited perturbation budget, *i.e.*, $\|\mathbf{\Delta x}\|_p \leq \epsilon$, in the input space.

In light of all these considerations, we turn to the "dual" optimization problem and opt to maximize the prediction loss of $g$ with a $\gamma\times$ "decayed" intermediate-level perturbation (*i.e.*, $(h(\mathbf{x}+\mathbf{\Delta x})-h(\mathbf{x}))/\gamma$ which is regarded as the directional guide $\mathbf{v}$). That is,

$$\max_{\|\mathbf{\Delta x}\|_p \leq \epsilon} L(g(h(\mathbf{x})+\frac{1}{\gamma}(h(\mathbf{x}+\mathbf{\Delta x})-h(\mathbf{x}))),y) = \max_{\|\mathbf{\Delta x}\|_p \leq \epsilon} L(g(\frac{1}{\gamma}h(\mathbf{x}+\mathbf{\Delta x})+(1-\frac{1}{\gamma})h(\mathbf{x})),y). \tag{3}$$

Since the method seems decays the intermediate-level perturbation from the benign features, we call it *intermediate-level perturbation decay (ILPD)*. Apparently, setting $\gamma = 1$ means attack without the perturbation modification and it makes the method equivalent to the baseline multi-step attack. While, enforcing $\gamma > 1$ leads to decayed perturbation as desired. The operation of $h(\mathbf{x}+\mathbf{\Delta x})/\gamma + (\gamma-1)h(\mathbf{x})/\gamma)$ resembles the mixup augmentation [51], except there is no randomness for $\gamma$ and $\gamma$ is suggested to be relatively large to encourage larger perturbation magnitude and larger loss.

Figure 2 demonstrates the performance of our ILPD, together with that of ILA. Solid dots $\bullet, \bullet, \ldots, \bullet$ represent achievable performance of our ILPD with $1/\gamma \in \{1.0, 0.9, \ldots, 0.1\}$, while the blue cross mark ✖ represents the performance of ILA in the figure. From the vertical axis, it can be seen that the ILPD adversarial examples are capable of tricking the victim model into showing significantly larger prediction loss (4.08->4.54), despite possessing lower perturbation magnitude in the feature space, comparing with ILA examples. Given $\gamma$ in the concerned range, ILPD-obtained perturbation directions are very effective in the sense of being adversarial. See the shaded area which illustrates how the attack performance varies with the perturbation magnitude along with these directions in the middle layer.

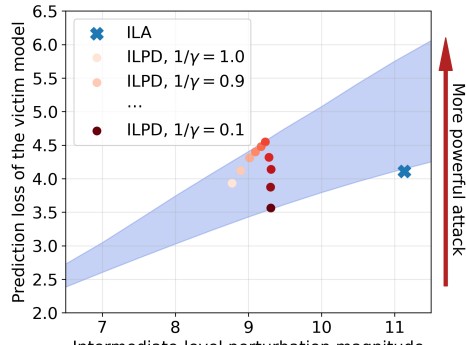

Figure 2: Comparing the achieved attack performance and the intermediate-level perturbation magnitude of ILA and our ILPD. Darker colors of the solid dots indicate larger values of $\gamma$. Best viewed in color.

It is worth noting that we normally suggest a relatively large $\gamma$ with $\gamma \geq 2$ (*i.e.*, $1/\gamma \leq 0.5$). The formulation of ILPD leads to a $\gamma\times$ larger perturbation magnitude anyway, than that of $(h(\mathbf{x}+\mathbf{\Delta x})-h(\mathbf{x}))/\gamma$ which is regarded as the directional guide $\mathbf{v}$. However, given $g$ being non-linear in practice, a $\gamma\times$ amplified intermediate-level perturbation does not truly lead to $\gamma\times$ larger prediction loss of $g$. In particular, the obtained loss is more difficult to be guaranteed especially when $\gamma$ is extremely large, *e.g.*, $\gamma \gg 10$ or, equivalently, $1/\gamma \ll 0.1$. Therefore, we simply apply with $0.1 \leq 1/\gamma \leq 0.5$ in our main experiments in Section 5. Comprehensive ablation studies are deferred to Section 5.4.

## 4 Analysis from the Gradient Alignment Perspective

Recall that the diversity in input gradients across models is the main obstacle to adversarial transferability, thus, in this section, we analyze ILPD from the perspective of gradient alignment between the substitute and victim models, in order to shed more light on its effectiveness.

### 4.1 Impacting Factors of Gradient Alignment

Aiming at compromising an unknown victim model $f_{\mathrm{t}}$, a baseline transfer-based attack (*e.g.*, I-FGSM) can generate adversarial example on a white-box substitute model $f$. In fact, if $\nabla_{\mathbf{x}}L_f(\mathbf{x}+\mathbf{\Delta x}_j,y) \approx$

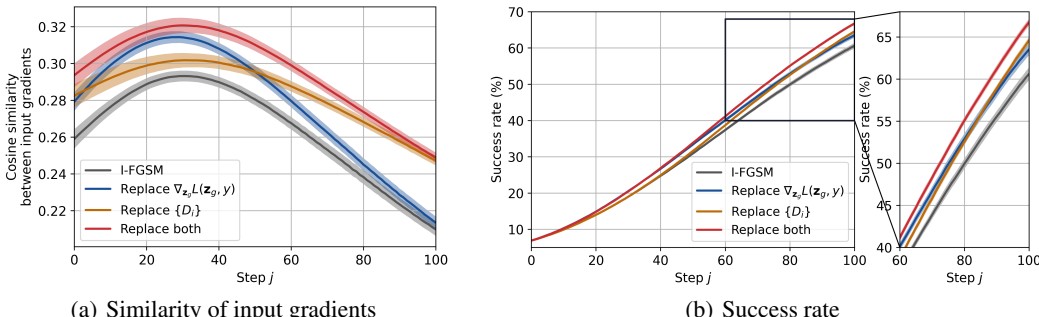

(a) Similarity of input gradients        (b) Success rate

Figure 3: How (a) the cosine similarity of input gradients and (b) success rate vary with I-FGSM iterations, using tens pairs of substitute and victim models. The shaded areas indicate the standard deviation of the pairs. Best viewed in color.

$\nabla_{\mathbf{x}} L_{f_t}(\mathbf{x} + \Delta \mathbf{x}_j, y)$, then the obtained adversarial examples will be as powerful on the victim model. Yet, the input gradients are often not well-aligned, and, as depicted in Figure 3(a), after some initial iterations, the input gradient of $L_f(\mathbf{x} + \Delta \mathbf{x}_j, y)$ and that of $L_{f_t}(\mathbf{x} + \Delta \mathbf{x}_j, y)$ may be increasingly different as $j$ increases, which is probably a major obstacle to the adversarial transferability of multi-step attacks.

Let us delve deep into this. For simplicity of notations, the activation functions $\phi_i$ is restricted to ReLU. Then, the input gradient $\nabla_{\mathbf{x}} L_f(\mathbf{x} + \Delta \mathbf{x}_j, y)$ can be written as the product of three terms:

$$\nabla_{\mathbf{x}} L_f(\mathbf{x} + \Delta \mathbf{x}_j, y) = \nabla_{\mathbf{x}} h(\mathbf{x} + \Delta \mathbf{x}_j) \nabla_{\mathbf{z}_h^{(j)}} L(g(\mathbf{z}_h^{(j)}), y)$$
$$= \nabla_{\mathbf{x}} h(\mathbf{x} + \Delta \mathbf{x}_j) \left( W_{\beta+1} \prod_{i=\beta+1}^{\alpha-1} D_i(\mathbf{z}_h^{(j)}) W_{i+1} \right) \nabla_{\mathbf{z}_g^{(j)}} L(\mathbf{z}_g^{(j)}, y), \quad (4)$$

following the chain rule, where $D_i(\mathbf{z}_h^{(j)})$ provides a binary matrix whose diagonal entry is set to 1 if it corresponds to a nonzero activation within the $i$-th layer, and it is set to 0 if otherwise. If there exist max pooling layers, they can be similarly formulated, and the gradient through these layers can also be given as a binary matrix in $\{D_i\}$.

The input gradient of a victim model $f_t$ (i.e., $\nabla_{\mathbf{x}} L_{f_t}(\mathbf{x} + \Delta \mathbf{x}_j, y)$) can be similarly formulated as in Eq. (4). The diversity between $\nabla_{\mathbf{x}} L_f(\mathbf{x} + \Delta \mathbf{x}_j, y)$ and $\nabla_{\mathbf{x}} L_{f_t}(\mathbf{x} + \Delta \mathbf{x}_j, y)$ probably comes from some difference in the activation masks in $\{D_i\}$ and/or diversity in the gradients of $L$ w.r.t. the logits (i.e., $\nabla_{\mathbf{z}_g} L(\mathbf{z}_g, y)$ where $j$ is dropped for simplicity of notation). Note that the gradients of $h$ are not concerned as it has been assumed in prior work that different models might share a similar $h$.

By replacing 1) activation masks in $\{D_i\}$ and/or 2) the gradient of $L$ w.r.t. the logits (i.e., $\nabla_{\mathbf{z}_g} L(\mathbf{z}_g, y)$) with those obtained on the victim model, we are capable of analyzing how much impact they actually make. An experiment is thus conducted, using ten pairs of substitute and victim models with the VGG-19 architecture. Figure 3 shows that the transferability can be remarkably improved by replacing any of them. Replacing $\nabla_{\mathbf{z}_g} L(\mathbf{z}_g, y)$ seems more effective at initial iterations while replacing the activation masks leads to slightly more improvement at the end. Given the cross-entropy loss which evaluates the discrepancy between the prediction probabilities and the one-hot ground-truth vector $\mathbf{y}$, we have $\nabla_{\mathbf{z}_g} L(\mathbf{z}_g, y) = \mathbf{p} - \mathbf{y}$ where $\mathbf{p} = s(\mathbf{z}_g)$ is a vector containing all prediction probabilities and $s$ is the softmax function. That said, at initial iterations, the difference in $\mathbf{p}$ between the models is a more influential factor, and, at later iterations, the difference in $\{D_i\}$ becomes dominant.

As the adversarial example is optimized on the substitute model, the prediction of a I-FGSM example is very likely to be less incorrect on the victim model even at the first iteration. Difference in $\mathbf{p}$ across models leads to difference in the input gradient, and it is possible that the difference become more and more severe, forming a vicious circle in some sense.

## 4.2 Analysis of ILPD and Some Other Attacks

Recall that the input gradient obtained by ILPD at the $j$-th iteration is

$$\nabla_{\mathbf{x}} h(\mathbf{x} + \Delta \mathbf{x}_j) \nabla_{\tilde{\mathbf{z}}_h^{(j)}} L(g(\tilde{\mathbf{z}}_h^{(j)}), y) \quad (5)$$

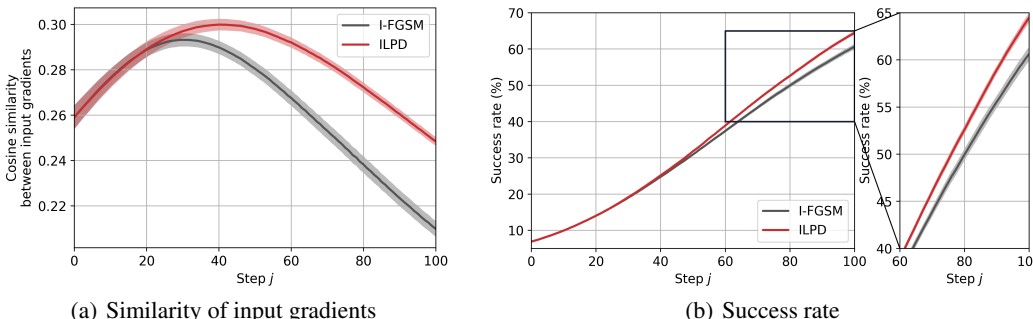

(a) Similarity of input gradients        (b) Success rate

Figure 4: How (a) the cosine similarity of input gradients and (b) success rate change along with I-FGSM and ILPD iterations, using ten pairs of substitute and victim models. Shaded areas indicate the standard deviation. Best viewed in color.

in which $\tilde{\mathbf{z}}_h^{(j)} = \frac{1}{\gamma}\mathbf{z}_h^{(j)} + (1 - \frac{1}{\gamma})\mathbf{z}_h^{(0)}$. It can be regarded as replacing $\nabla_{\mathbf{z}_h^{(j)}} L(g(\mathbf{z}_h^{(j)}), y)$ in Eq. (4) with the gradient of $L$ $w.r.t.$ to some less adversarial intermediate-level perturbation $\tilde{\mathbf{z}}_h^{(j)}$. Less adversarial intermediate-level perturbations leads to less incorrect predictions, which probably aligns better with that of the victim model.

Figure 4 shows that ILPD leads to more aligned input gradient across models, especially after the first 20 iterations. The attack success rates are compared in Figure 4(b), and it can be seen that, thanks to the improved gradient alignment, ILPD is capable of generating more transferable adversarial examples in comparison to I-FGSM, achieving $+4.00\%$ absolute gains at the 100-th iteration. More detailed analyses show that the ILPD examples lead to activation masks and prediction probabilities better aligned with those of the victim model.

In addition to our ILPD, some other intermediate-level methods can also be seen as beneficial to gradient alignment. In particular, for ILA [20], we have

$$\mathbf{v} = \mathbf{z}_h^{(u)} - \mathbf{z}_h^{(0)} \approx \frac{\|\mathbf{z}_h^{(u)} - \mathbf{z}_h^{(0)}\|_2}{L(g(\mathbf{z}_h^{(u)}), y) - L(g(\mathbf{z}_h^{(0)}), y)}\nabla_{\mathbf{z}_h^{(0)}} L(g(\mathbf{z}_h^{(0)}), y), \tag{6}$$

where the Taylor series is adopted to obtain the approximation. Therefore, optimizing Eq. (2) at the $j$-th step of an iterative optimizer can be considered as adopting the gradient $w.r.t.$ $\mathbf{z}_h^{(0)}$ as a replacement of the gradient $w.r.t.$ $\mathbf{z}_h^{(j)}$. Without injecting noise or augmenting data at each iteration, ILA++ [26, 16] and NAA [52] can also be approximately regarded as replacing the gradient $w.r.t.$ $\mathbf{z}_h^{(j)}$ with the gradient $w.r.t.$ $\mathbf{z}_h^{(0)}$.

In addition to these efforts, Guo $et\,al.$ proposed LinBP [15] which back-propagates linearly in later layers. It can be regarded as replacing each $D_i$ with an identity matrix for computing input gradients. Though effective, it mainly takes care of $\{D_i\}$ and somehow ignores the difference of $\nabla_{\mathbf{z}_g} L(\mathbf{z}_g, y)$ across models.

## 5 Experiments

Experimental settings will be introduced in Section 5.1. We will compare our solution with a variety of state-of-the-arts to evaluate its effectiveness in Section 5.2. We combine our solution with some existing methods to further improve the adversarial transferability in Section 5.3, and we perform some ablation studies in Section 5.4.

### 5.1 Experimental Settings

Our experiments were conducted on CIFAR-10 [24] and ImageNet [35] under $\ell_\infty$ constraint in the black-box setting. We focus on untargeted attacks and use I-FGSM attack as the back-end method, just like in previous work [46, 11, 28, 42]. We set the perturbation budget to $\epsilon = 4/255$ and $8/255$ for attacks on CIFAR-10 and ImageNet, respectively. Further, on CIFAR-10, we used a VGG-19 with batch normalization [36] as the substitute model, and a *different* VGG-19 with batch normalization,

Table 1: Transfer-based attack performance on ImageNet. The attacks are performed under the $\ell_\infty$ constraint with $\epsilon = 8/255$ in the untargeted setting.

| Method | ResNet -50 | ResNet -152 | DenseNet -121 | WRN -101 | Inception v3 | RepVGG -B1 | VGG -19 | Mixer -B | ConvNeXt -B | ViT -B | DeiT -B | Swin B | BEiT -B | Average |
|---|---|---|---|---|---|---|---|---|---|---|---|---|---|---|
| I-FGSM | 38.20% | 18.88% | 49.24% | 47.62% | 22.14% | 33.54% | 44.98% | 9.00% | 9.68% | 4.42% | 3.96% | 5.16% | 3.84% | 22.36% |
| MI-FGSM | 47.88% | 25.98% | 59.96% | 56.26% | 30.42% | 43.22% | 54.28% | 13.36% | 13.18% | 6.74% | 5.80% | 7.68% | 6.62% | 28.57% |
| DI$^2$-FGSM | 74.24% | 47.20% | 90.28% | 84.76% | 50.84% | 77.44% | 83.50% | 18.62% | 29.76% | 12.82% | 11.96% | 13.92% | 14.12% | 46.88% |
| TI-FGSM | 38.72% | 19.72% | 49.60% | 49.00% | 22.72% | 33.90% | 45.86% | 9.36% | 9.20% | 4.70% | 3.94% | 5.12% | 4.08% | 22.76% |
| SI-FGSM | 51.46% | 28.62% | 71.42% | 65.82% | 36.50% | 47.60% | 59.18% | 12.42% | 12.22% | 6.34% | 5.66% | 6.28% | 6.48% | 31.54% |
| SGM | 71.46% | 48.72% | 73.28% | 74.26% | 35.86% | 65.60% | 75.10% | 19.08% | 22.84% | 10.66% | 8.64% | 12.26% | 9.96% | 40.59% |
| LinBP | 73.00% | 52.86% | 81.56% | 78.18% | 39.40% | 68.28% | 81.54% | 13.90% | 15.80% | 7.70% | 5.82% | 8.90% | 7.98% | 41.15% |
| Admix | 63.52% | 37.64% | 71.94% | 73.68% | 31.84% | 60.50% | 70.92% | 11.90% | 15.14% | 6.62% | 5.10% | 7.62% | 6.24% | 35.59% |
| TAIG-R | 66.26% | 44.28% | 87.50% | 84.46% | 59.32% | 72.36% | 80.94% | 18.86% | 14.54% | 10.96% | 8.00% | 9.56% | 13.46% | 43.88% |
| NAA | 74.14% | 58.84% | 86.16% | 84.34% | 57.40% | 76.80% | 82.96% | 20.96% | 19.86% | 9.92% | 8.42% | 11.56% | 11.36% | 46.36% |
| ILA++ | 75.28% | 55.56% | 84.78% | 83.46% | 50.98% | 72.48% | 78.48% | 22.22% | 30.28% | 13.76% | 12.18% | 16.50% | 14.94% | 46.99% |
| ILPD | **83.96%** | **69.86%** | **90.68%** | **90.28%** | **64.70%** | **85.90%** | **88.10%** | **29.82%** | **44.54%** | **20.42%** | **21.52%** | **26.88%** | **25.12%** | **57.06%** |

a ResNet-18 [18], a PyramidNet [17], GDAS [10], a WRN-28-10 [50], a ResNeXt-29 [47], and a DenseNet-BC [19] [2] as the victim models, following some prior work [15, 26, 16]. All victim models (except for the VGG-19 which was trained by ourselves) were pre-trained by others and directly collected for experiments. On ImageNet, we used a ResNet-50 [18] model as the substitute model, and collect 13 victim models, including a *different* ResNet-50 [18], a VGG-19 [36], a ResNet-152 [18], a Inception v3 [37], a DenseNet-121 [19], a WRN-101 [50], a RepVGG-B1 [9], a MLP Mixer-B [38], a ConvNeXt-B [31], a ViT-B [13], a DeiT-B [39], a Swin-B [30], and a BEiT-B [3] [3].

The image size of adversarial examples was $224 \times 224$ on ImageNet and $32 \times 32$ on CIFAR-10, without crop, following prior work [28, 42]. We performed adversarial attacks on all test data in CIFAR-10 and 5000 randomly sampled examples from the ImageNet validation data. The adversarial examples obtained at each iteration of the attack should be clipped to [0, 1], in order to ensure that only valid images are fed to DNNs. We run 100 iterations with a step size of $1/255$ for all attack methods on both CIFAR-10 and ImageNet in this section, and we set a smaller step size of $\epsilon/100$ for Section 4 as we care more about the trend of overfitting before exhausting the perturbation budget. ILPD was performed at the output of the fourth VGG block for VGG-19 on CIFAR-10 and the output of the last building block of the second ResNet meta layer for ResNet-50 on ImageNet, with $\gamma$ tuned in the range satisfying $0.1 \leq 1/\gamma \leq 0.5$. The advanced version of ILA++ [16] and NAA [52] are adopted as competitors. Since both methods inject some random transformations to the benign examples, when computing gradients, we also add Gaussian noise with a standard deviation of 0.05 to the benign examples at each iteration, for fair comparison. For competitors, we directly adopt their official implementations and hyper-parameters, and we attack on the same CIFAR-10 and ImageNet images for all methods. All experiments are performed on an NVIDIA V100 GPU.

## 5.2 Comparison with State-of-the-arts

The comparison results are summarized in Table 1 and 2. For ImageNet, 13 victim models, including four vision transformers, one multi-layer perceptron model, and a model that has the same architecture of the substitute model, are adopted. For CIFAR-10, 7 victim models are adopted. We compare our method with a gradient stabilization method (*i.e.*, MI-FGSM [11]), some input augmentation methods (*i.e.*, DI$^2$-FGSM [46], TI-FGSM [12], SI-FGSM [28], and the recent Admix [42]), compelling intermediate-level attacks (*i.e.*, ILA++ [16] and NAA [52]), some back-propagation modification methods (*i.e.*, SGM [45] and LinBP [15]), and a very recent method called TAIG [21].

Experiment results in the tables show that our method outperforms all these competitors by large margins. Specifically, it can be seen that with the assistance of our ILPD, it is possible to achieve an average success rate of **57.06%** on ImageNet, which outperforms the second best (*i.e.*, ILA++) by **+10.07%**. On CIFAR-10, we achieve **62.82%**, which outperforms the second best by **+3.88%**.

Attacking some newly designed models is challenging when using ResNet as the substitute model. The I-FGSM baseline can only achieve a success rate of less than $10\%$ against these models (see the right half of Table 1) and even $< 5\%$ against some vision transformers. However, our method

---

[2]https://github.com/bearpaw/pytorch-classification
[3]https://github.com/rwightman/pytorch-image-models

Table 2: Transfer-based attack performance on CIFAR-10. The attacks are performed under the $\ell_\infty$ constraint with $\epsilon = 4/255$ in the untargeted setting.

| Method | VGG-19 | ResNet-18 | WRN | ResNeXt | DenseNet | PyramidNet | GDAS | Average |
|---|---|---|---|---|---|---|---|---|
| I-FGSM | 71.18% | 67.67% | 51.28% | 52.77% | 48.74% | 12.44% | 36.23% | 48.62% |
| MI-FGSM | 73.56% | 69.65% | 53.55% | 55.11% | 50.36% | 13.03% | 38.07% | 50.48% |
| DI$^2$-FGSM | 77.40% | 72.53% | 56.68% | 57.58% | 53.68% | 15.78% | 40.70% | 53.48% |
| TI-FGSM | 71.32% | 67.67% | 51.53% | 52.76% | 48.63% | 12.43% | 36.44% | 48.68% |
| SI-FGSM | 70.51% | 72.70% | 55.73% | 57.51% | 53.65% | 14.19% | 39.77% | 52.01% |
| LinBP | 78.19% | 75.08% | 59.35% | 61.92% | 57.89% | 16.88% | 43.56% | 56.12% |
| Admix | 78.17% | 74.51% | 58.18% | 59.94% | 55.69% | 15.05% | 42.42% | 54.85% |
| TAIG-R | 73.72% | 71.25% | 51.57% | 52.72% | 50.01% | 12.93% | 39.64% | 50.26% |
| NAA | 71.16% | 71.00% | 56.62% | 58.67% | 56.68% | 18.31% | 46.25% | 54.10% |
| ILA++ | 79.77% | 78.81% | 63.36% | 63.91% | 60.38% | 18.86% | 47.51% | 58.94% |
| ILPD | **83.52%** | **81.63%** | **68.03%** | **67.97%** | **65.37%** | **19.44%** | **53.79%** | **62.82%** |

achieves impressive improvements in attacking these models. For instance, compared with the I-FGSM baseline, when attacking the MLP-Mixer and ConvNeXt models, our ILPD obtains $+20.82\%$ and $+34.86\%$ absolute gains, respectively, and when attacking the vision transformers, our ILPD obtains an absolute gain of $+19.14\%$.

**Attack robust models.** We also tested our method on the task of attacking robust models. We collected a robust Inception v3 and a robust EfficientNet-B0 from the timm [44] repository, and obtain a robust ResNet-50 and a robust DeiT-S [40] from Bai *et al.* [2]'s open-source repository [4] as the victim models. Adversarial examples generated on the ResNet-50 substitute model were applied for this experiment. Our ILPD is compared with

Table 3: Performance in attacking robust ImageNet models. The attacks are performed under the $\ell_\infty$ constraint with $\epsilon = 8/255$ in the untargeted setting.

| Method | Inception v3 (robust) | EfficientNet (robust) | ResNet-50 (robust) | DeiT-S (robust) | Average |
|---|---|---|---|---|---|
| I-FGSM | 11.68% | 10.88% | 9.10% | 10.56% | 10.56% |
| DI$^2$-FGSM | 15.84% | 34.04% | 9.66% | 11.02% | 17.64% |
| NAA | 22.86% | 37.02% | 10.80% | 11.42% | 20.53% |
| ILA++ | 18.22% | 34.62% | 10.46% | 11.40% | 18.68% |
| ILPD | **25.30%** | **50.02%** | **11.42%** | **11.90%** | **24.66%** |

DI$^2$-FGSM, NAA, and ILA++, which are the most competitive methods in Table 1, and we summarize the results in Table 3. It shows that our method outperforms competitors in attacking these robust models and getting $+4.13\%$ improvement, compared with NAA, which is the second best method. In addition to the robust models, we also test our method on two advanced defense methods, *i.e.*, random smoothing [7] and NRP [33]. Specifically, our ILPD achieves a success rate of $12.90\%$ in attacking a smoothed ResNet-50 on ImageNet, while the second best method (NAA) and the baseline show $11.84\%$ and $7.20\%$, respectively. For NRP, the adversarial examples generated by our method can achieve an average success rate of $20.03\%$ after purification. Meanwhile, the second best method (DI$^2$-FGSM) achieves $14.54\%$, and the baseline achieves $9.39\%$.

## 5.3 Combination with Existing Methods

Our method can be readily combined with other attacks. In this section, we evaluate the combination performance. We tried combining our method with different sorts of attacks, including a gradient stabilization method, *i.e.*, MI-FGSM [11] and two input augmentation methods, *i.e.*, DI$^2$-FGSM [46] and Admix [52]. The results are reported in Table 4. It shows that the adversarial transferability can indeed be further improved by combining our method with these prior arts. Specifically, when combining with DI$^2$-FGSM, the best average success rate, *i.e.*, **63.23%**, is obtained.

Table 4: Combination of our ILPD with MI-FGSM, DI$^2$-FGSM, and Admix on ImageNet. The attacks use I-FGSM as the back-end attack and are performed under the $\ell_\infty$ constraint with $\epsilon = 8/255$ in the untargeted setting.

| Method | ResNet -50 | ResNet -152 | DenseNet -121 | WRN -101 | Inception v3 | RepVGG -B1 | VGG -19 | Mixer -B | ConvNeXt -B | ViT -B | DeiT B | Swin -B | BEiT -B | Average |
|---|---|---|---|---|---|---|---|---|---|---|---|---|---|---|
| MI-FGSM | 47.88% | 25.98% | 59.96% | 56.26% | 30.42% | 43.22% | 54.28% | 13.36% | 13.18% | 6.74% | 5.80% | 7.68% | 6.62% | 28.57% |
| +ILPD | **84.90%** | **72.22%** | **91.74%** | **91.42%** | **69.44%** | **87.66%** | **89.12%** | **34.62%** | **48.00%** | **24.20%** | **25.36%** | **29.16%** | **30.02%** | **59.84%** |
| DI$^2$-FGSM | 74.24% | 47.20% | 90.28% | 84.76% | 50.84% | 77.44% | 83.50% | 18.62% | 29.76% | 12.82% | 11.96% | 13.92% | 14.12% | 46.88% |
| +ILPD | **85.26%** | **74.62%** | **92.40%** | **91.36%** | **74.42%** | **88.74%** | **90.10%** | **37.98%** | **53.42%** | **29.50%** | **31.22%** | **35.32%** | **37.70%** | **63.23%** |
| Admix | 63.52% | 37.64% | 71.94% | 73.68% | 31.84% | 60.50% | 70.92% | 11.90% | 15.14% | 6.62% | 5.10% | 7.62% | 6.24% | 35.59% |
| +ILPD | **86.08%** | **75.48%** | **92.48%** | **91.66%** | **70.44%** | **88.84%** | **90.52%** | **33.42%** | **47.94%** | **23.24%** | **24.12%** | **29.66%** | **29.78%** | **60.28%** |

[4]https://github.com/ytongbai/ViTs-vs-CNNs

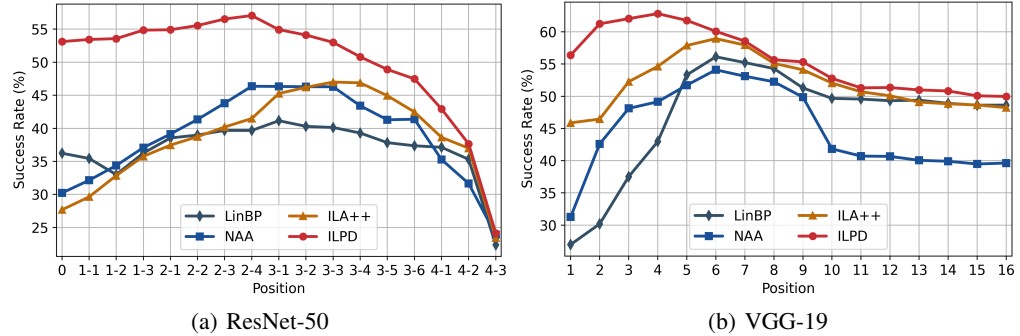

(a) ResNet-50                (b) VGG-19

Figure 5: How the average success rate change with different choices of position on (a) ImageNet trained ResNet-50 and (b) CIFAR-10 trained VGG-19. For the ResNet-50 model, "0" represent the stem layer and "1-1" represent the first building block of the first ResNet meta layer.

## 5.4 Ablation Study

**The effect of position.** As an intermediate-level attack, our method needs to choose a position to split $f$ in $h$ and $g$. Such a position selection affects the performance of all intermediate-level attacks. We vary the position and compare our method with ILA++ [16], NAA [52], and LinBP [15] in Figure 5. The performance on ImageNet and CIFAR-10 are both tested, using ResNet-50 and VGG-19 as the substitute models, respectively. Apparently, our ILPD demonstrates consistently higher adversarial transferability on all choices of the position. In particular, our method significantly outperforms ILA++, NAA, and LinBP when choosing a lower layer to split $f$ into $h$ and $g$. When using ResNet-50 as the substitute model, choosing any of the first eleven layers (*i.e.*, from "0" to "3-3") can achieve an average success rate of more than $53\%$ on ImageNet, even simply choosing the first layer will lead to an average success rate of $53.12\%$, which is already superior to all competitors in Table 1.

**The effect of $\gamma$.** We evaluate ILPD with varying $\gamma$ on ImageNet and CIFAR-10 in Figure 6. The victim models are the same as the ones in Table 1 and Table 2. It can be seen that using $1/\gamma < 1$ always leads to improved performance compared with the baseline (which is equivalent to adopting $1/\gamma = 1$). On ImageNet, the optimal performance is obtained with $1/\gamma = 0.1$ (*i.e.*, $\gamma = 10$) and ResNet-50, while, for CIFAR-10, $1/\gamma = 0.3$ (*i.e.*, $\gamma = 10/3$) is more effective. That is, on ImageNet, we aim to maximize the prediction loss of a perturbation which is cut by

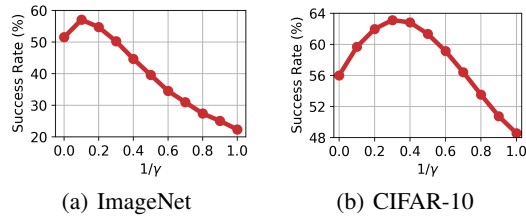

(a) ImageNet       (b) CIFAR-10

Figure 6: How the average success rate changes with $\gamma$ using (a) ImageNet trained ResNet-50 and (b) CIFAR-10 trained VGG-19 as the substitute models.

$10\times$ in the middle layer. It is possible that employing $\gamma > 10$ leads to even better performance on ImageNet, yet we do not aim to carefully tune $\gamma$ and would like to leave it to future work.

## 6 Conclusion

In this paper, we have revisited some intermediate-level attacks and shown that, limited by their two-stage training scheme, the obtained intermediate-level perturbations will deviate from the directional guides and such a deviation largely impairs the attack performance of the crafted adversarial examples. In order to overcome this limitation, we have developed a method that performs attacks through a single stage of optimization and guarantees the obtained intermediate-level perturbation possesses a larger magnitude when compared with a directional guide and exactly follows its direction that can effectively increase the prediction loss of the model. The developed method is formed as adopting intermediate-level perturbation decay when computing gradients. Extensive experimental results on ImageNet and CIFAR-10 have shown that our method outperforms state-of-the-arts by large margins in attacking various victim models, including convolutional models, multi-layer perceptron model, vision transformers, and robust models. Moreover, we have shown that our method can be easily combined with prior methods to craft more transferable adversarial examples.

# 7 Acknowledgment

This material is based upon work supported by the National Science Foundation under Grant No. 1801751 and 1956364.

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

# A Experiment Results of Using Other Substitute Models

We compare our method with other intermediate-level attacks using various substitute model on ImageNet. The comparison results are summarized in Table 5. Three vision transformers and one multi-layer perceptron model are considered as the substitute models, including a ViT-B [13], a DeiT-B [39], a Swin-B [30], and a MLP Mixer-B [38]. The victim models are the same as those in Table 1. For ILPD, $h$ is always split to be the first two blocks of these models and we fix $1/\gamma = 0.1$. For NAA and ILA++, we tested the results of all possible intermediate layer selections and report the best results among them. The average success rates are compared as follows, and the victim models are the same as those in Table 1 of our paper. Experiment results in the table show that our method outperforms all these competitors by large margins.

Table 5: Transfer-based attack performance of using different substitute models on ImageNet. The attacks are performed under the $\ell_\infty$ constraint with $\epsilon = 8/255$ in the untargeted setting. "Average" is obtained on victim models different from the substitute model.

| Substitute | Method | ResNet -50 | ResNet -152 | DenseNet -121 | WRN -101 | Inception v3 | RepVGG -B1 | VGG -19 | Mixer -B | ConvNeXt -B | ViT -B | DeiT -B | Swin B | BEiT -B | Average |
|---|---|---|---|---|---|---|---|---|---|---|---|---|---|---|---|
| ViT-B | ILA++ | 22.12% | 17.44% | 28.20% | 21.28% | 28.52% | 29.30% | 32.22% | 47.06% | 19.00% | 97.22% | 40.12% | 29.50% | 43.34% | 29.84% |
| | NAA | 31.90% | 27.64% | **45.74%** | 35.36% | **46.56%** | **44.98%** | **46.36%** | 58.32% | 26.76% | 94.58% | 60.62% | 40.52% | 57.52% | 43.52% |
| | ILPD | **34.96%** | **32.64%** | 43.78% | **37.38%** | 44.02% | 44.60% | 44.36% | **58.82%** | 36.38% | 96.42% | **61.92%** | **54.16%** | **66.54%** | **46.63%** |
| DeiT-B | ILA++ | 20.92% | 14.40% | 24.76% | 18.60% | 25.24% | 29.46% | 33.00% | 33.94% | 18.24% | 27.22% | **87.54%** | 18.38% | 19.72% | 23.66% |
| | NAA | 29.74% | 25.04% | 36.52% | 30.26% | 38.26% | 39.18% | 39.42% | 45.78% | 31.38% | 39.42% | 75.32% | 31.54% | 37.62% | 35.35% |
| | ILPD | **40.14%** | **36.20%** | **42.02%** | **36.80%** | **40.82%** | **44.80%** | **43.76%** | **53.34%** | **45.68%** | **51.82%** | 81.74% | **46.36%** | **49.64%** | **44.28%** |
| Mixer-B | ILA++ | 22.10% | 14.46% | 24.52% | 19.08% | 27.96% | 25.48% | 27.64% | 96.64% | 15.08% | 22.76% | 23.76% | 12.68% | 14.24% | 20.81% |
| | NAA | 33.28% | 23.24% | **42.92%** | 28.86% | 41.56% | 41.48% | **43.88%** | **92.62%** | 26.54% | 42.76% | 52.88% | 24.64% | 29.06% | 35.93% |
| | ILPD | **39.84%** | **30.84%** | 42.56% | **34.40%** | **41.88%** | **44.44%** | 42.56% | 91.56% | **40.80%** | **49.88%** | **58.26%** | **36.38%** | **38.74%** | **41.72%** |
| Swin-B | ILA++ | 12.68% | 8.36% | 13.44% | 9.22% | 14.30% | 18.92% | 22.78% | 12.64% | 19.84% | 9.14% | 7.64% | 83.96% | 9.80% | 13.23% |
| | NAA | 16.36% | 13.10% | 22.72% | 16.54% | 22.90% | 26.78% | 28.84% | 22.04% | 28.80% | 17.70% | 19.94% | 71.90% | 21.16% | 21.41% |
| | ILPD | **31.04%** | **27.44%** | **32.66%** | **25.88%** | **28.06%** | **43.08%** | **42.52%** | **32.06%** | **55.48%** | **40.14%** | **35.42%** | 89.56% | **47.42%** | **36.77%** |

