# OpenReview forum: "Improving Adversarial Transferability via Intermediate-level Perturbation Decay"
_NeurIPS.cc/2023/Conference — NeurIPS 2023 poster_

### Official Review · Reviewer_ektX · 2023-06-28

**Soundness:** 3 good
**Presentation:** 3 good
**Contribution:** 3 good
**Rating:** 6
**Confidence:** 5

**Summary:**

The paper addresses the limitations of intermediate-level attacks, highlighting how their two-stage training scheme leads to perturbations that deviate from directional guides, significantly impairing attack performance. To overcome this limitation, the authors propose a single-stage optimization method that ensures intermediate-level perturbations have larger magnitudes and precisely follow the directional guides, effectively increasing model prediction loss. Extensive experiments on ImageNet and CIFAR-10 demonstrate that the proposed method outperforms existing techniques when attacking various victim models. Furthermore, the paper shows that the proposed method can be easily combined with prior approaches to craft more transferable adversarial examples.






**Strengths:**

This paper recognizes the limitations of current intermediate-level attacks and proposes a novel method named ILPD. This method crafts adversarial examples using a single stage of optimization, encouraging the intermediate-level perturbation to be both in an effective adversarial direction and possess a significant magnitude simultaneously. A large number of experiments demonstrate that ILPD outperforms SOTA methods by large margins.

**Weaknesses:**

The method seems sensitive to $\gamma$ and the intermediate layer to split h and g.

**Questions:**

1 Line 200-201. It is reasonable that the gradients of $h$ are not concerned, but why $W_{\beta+1}$ (after the intermediate layer) not considered?

2 If change the surrogate model, do you need to adjust the intermediate position and $\gamma$?

**Limitations:**

The method appears to be sensitive to the parameters $\gamma$ and the choice of intermediate layer for splitting h and g. However, in real-world scenarios, it is often impractical to tune these parameters effectively based on the victim models.

---

> ### Author Rebuttal · Authors · 2023-08-09
>
> Thanks for your positive feedback. Our response to the comments is provided as follows.
> &nbsp;
>
> > The method seems sensitive to $\gamma$ and the intermediate layer to split $h$ and $g$.
>
> **Response:** Just like all other intermediate-level attacks, although our ILPD needs to tune the position to achieve its optimal performance, we can observe from our discussions in Section 5.4 that our method is less sensitive to the choice of position compared to other methods. Moreover, accordingly, to experiments in the ILA paper, the optimal choice of position can actually be found on the substitute model itself thus it is generally not a concern. As for $1/\gamma$, Figure 6 in the paper shows that our method outperforms all its competitors with a wide range of $1/\gamma$ ($[0, 0.3]$ and $[0.1, 0.6]$ on ImageNet and CIFAR-10, respectively), showing that the choice of $1/\gamma$ is not a big concern to ensure its superiority over state-of-the-arts.
> &nbsp;
>
> > Line 200-201. It is reasonable that the gradients of $h$ are not concerned, but why $W_{\beta + 1}$ (after the intermediate layer) not considered?
>
> **Response:** We appreciate the comment. Indeed, the difference in $W_{\beta + 1}$ between the substitute model and the victim models also affects the transferability. However, such a gap cannot be easily bridged, unless we modify the parameters of the substitute model to make it more similar to the victim parameters, which seems infeasible from our perspective. By contrast, the difference in activation masks in $\{D_i\}$ and/or diversity in the gradients of $L$ _w.r.t._ the logits can be somehow reduced by modifying the intermediate-level representations on the substitute model, thus we mainly discuss them in our paper.
> &nbsp;
>
> > If change the surrogate model, do you need to adjust the intermediate position and $\gamma$?
>
> **Response:** To show whether the same hyper-parameters work on different substitute models, we conduct an experiment on ImageNet to compare the performance of ResNet-50, ViT-B, DeiT-B, Swin-B, or the MLP Mixer-B as the substitute model. $h$ is always split to be the first two blocks of these models and we fix $1/\gamma=0.1$ for them. For NAA and ILA++, we tested the results of all possible intermediate layer selections and report the best results among them. The average success rates are shown as follows and results are compared with the I-FGSM baseline, NAA, and ILA++. The victim models are the same as those in Table 1 in our paper. It can be seen that the obvious superiority of our ILPD holds.
>
> | Substitute model | NAA | ILA++ | ILPD |
> |:------------------:|:--------:|:-----:|:-------:|
> | ResNet-50        |   46.36%     |  46.99%   |   **57.06%**    |
> | ViT-B                   |        43.52%      |      35.02%    |       **46.63%**    |
> | DeiT-B                 |        35.35%      |     23.66%     |      **44.28%**     |
> | Swin-B                |          21.41%     |      13.23%     |     **36.77%**       |
> | MLP Mixer-B       |       33.18%       |     20.81%    |      **41.72%**      |

---

> > ### Comment · Reviewer_ektX · 2023-08-18
> > **Thanks for the response**
> >
> > The authors have addressed my question. I intend to maintain my current rating.

---

> > > ### Author Response · Authors · 2023-08-18
> > > **Thanks to the reviewer**
> > >
> > > Dear Reviewer ektX,
> > >
> > > Thanks for responding to our rebuttal. We are glad to know that your questions have been addressed!
> > >
> > > Best regards,
> > > Authors

---

### Official Review · Reviewer_maPS · 2023-07-03

**Soundness:** 2 fair
**Presentation:** 2 fair
**Contribution:** 2 fair
**Rating:** 5
**Confidence:** 4

**Summary:**

In this paper, the authors propose an intermediate-level attack method with a single stage of optimization to improve attack transferability. When computing gradients, it adopts intermediate-level perturbation decay to process a larger magnitude. The experiment validations on the ImageNet and CIFAR-10 datasets are given to demonstrate its effectiveness to improve the attack transferability.

**Strengths:**

1. The proposed method presents a single stage with a larger magnitude rather than two stage training.
2. The proposed method is a pluggable module that can be deployed in other attack methods.
3. The experiments are thorough with various victim models, including CNNs, MLP, vision transformers, and robust models.


**Weaknesses:**

1. From Figure 1, there is no obvious difference between the victim model and the substitute model. The reviewer is confused that these two figures are less related to the statement in Section 3.1.
2. The construction of this paper is disordered. The authors introduce the existing intermediate-level attack method with the proposed method, which covers the contribution of the proposed method. The authors should highlight its own contribution.
3. For the experiments, the authors only use the ResNet-50 model as the substitute model to test the attack transferability on other models. However, a cross-model evaluation could be better, i.e., using MLP or transformers to test on other types of models.
4. The writing of this paper needs to be improved.
- For the introduction, the authors do not provide a brief introduction of the proposed method.
- For the related work, the white-box attack is less related to this paper. And existing intermediate-level attack methods are missed.
- There are also some typos.
a) In line 31, be discuss -> be discussed.
b) In line 327, there exists two duplicate ‘to all competitors’.


**Questions:**

1. Figure 1 does not give an intuitive visualization about the problem of existing intermediate-level attacks. The motivation of this paper should be stated clearly.
2. Lack of related works with existing intermediate-level attack methods. The whole construction of this paper should be improved.
3. How about the transferability with other types of models as the substitute model?
4. Fix the typos and polish this paper again.


**Limitations:**

Limitations not clearly elaborated.

---

> ### Author Rebuttal · Authors · 2023-08-09
>
> Thanks for your feedback. Our responses to the comments are provided as follows.
> &nbsp;
>
> > From Figure 1, there is no obvious difference between the victim model and the substitute model. The reviewer is confused that these two figures are less related to the statement in Section 3.1.
>
> **Response:** Figure 1 is given to show the negative impact of the directional deviation from the guide in middle layers. In Figure 1, we can observe that with a similar scalar projection, more severe deviation leads to lower prediction loss on both the substitute model (Figure 1(a)) and the victim model (Figure 1(b)). We use blue and orange colors to indicate different ranges of prediction loss in the two subfigures, and it can be seen that lower prediction loss is generally achieved on the victim model.
>
> The intermediate-level perturbation obtained by ILA is marked by the purple box in the figure. The figure further demonstrates that ILA leads to directional deviation from the guide (as its x-axis value, _i.e._, the cosine similarity, is smaller than 1.0), and, in particular, if such a deviation can be addressed, higher attack performance can be achieved by adversarial examples whose intermediate-level perturbations are even smaller (as darker colors, _i.e._, higher prediction loss, can be achieved with points on the y-axis whose scalar projections are even smaller than that of the ILA point). We have discussed these points in detail in the last paragraph in Section 3.1.
> &nbsp;
>
> > The construction of this paper is disordered. The authors introduce the existing intermediate-level attack method with the proposed method, which covers the contribution of the proposed method. The authors should highlight its own contribution.
>
> **Response:** Section 3.1 not only introduces previous intermediate-level attacks, but also discusses and analyzes their limitations which inspire our work, _i.e._, ILPD. We believe such a discussion is also part of our contributions and it will inspire future work in this field. Moreover, without such a gentle introduction and discussion, it will be more difficult to capture the motivation of our method. We are more than glad to highlight more about the contribution of this paper in our revision.
> &nbsp;
>
> > For the experiments, the authors only use the ResNet-50 model as the substitute model to test the attack transferability on other models. However, a cross-model evaluation could be better, i.e., using MLP or transformers to test on other types of models.
>
> **Response:** We appreciate the suggestion and have done such a comparison on ImageNet, by using ViT-B, DeiT-B, Swin-B, and MLP Mixer-B as the substitute models. $h$ is always split to be the first two blocks of these models and we fix $1/\gamma=0.1$. For NAA and ILA++, we tested the results of all possible intermediate layer selections and report the best results among them. The average success rates are compared as follows, and the victim models are the same as those in Table 1 of our paper.
>
> | Substitute model | NAA | ILA++ | ILPD |
> |:------------------:|:-----:|:-------:|:-------:|
> | ViT-B                   |        43.52%      |      35.02%    |       **46.63%**    |
> | DeiT-B                 |        35.35%      |     23.66%     |      **44.28%**     |
> | Swin-B                |          21.41%     |      13.23%     |     **36.77%**       |
> | MLP Mixer-B       |       33.18%       |     20.81%    |      **41.72%**      |
> &nbsp;
>
> > For the introduction, the authors do not provide a brief introduction of the proposed method.
>
> **Response:** As has been discussed in the paper, since the name of the proposed method itself contains much information, we have briefly introduced our method in the introduction section as "In this paper, we propose a method that encourages the intermediate-level perturbation to possess a greater magnitude than a directional guide by its nature and to be in the same adversarial direction as that of the guide. This is achieved by introducing intermediate-level perturbation decay (ILPD) in a single stage of optimization." We are more than glad to introduce more in an updated version of the paper.
> &nbsp;
>
> > For the related work, the white-box attack is less related to this paper. And existing intermediate-level attack methods are missed.
>
> **Response:** Since I-FGSM is considered as the baseline and it was first introduced as a white-box method, we briefly introduce white-box attacks in Section 2. Notations are also given during such a brief introduction. Following your suggestion, we will consider revising Section 2 to further simplify such an introduction, and we will perform a more comprehensive literature review in this section.
> &nbsp;
>
> > There are also some typos. a) In line 31, be discuss -> be discussed. b) In line 327, there exists two duplicate ‘to all competitors’.
>
> **Response:** The authors would like to thank the reviewer for pointing out typos. All of them will be addressed in an updated version of the paper.

---

> > ### Comment · Reviewer_maPS · 2023-08-14
> >
> > Thanks for the response to my questions.
> >
> > The authors clarified some of my concerns, especially for more supportive experiments. However,  Figure 1 is still hard to understand for me. I think the authors can solve the remaining questions in this rebuttal and I am willing to upgrade my score.
> >
> > I hope the authors will provide the complete results in the revised paper and polish this paper again.

---

> > > ### Author Response · Authors · 2023-08-14
> > > **Thanks to the reviewer**
> > >
> > > It's great to know that your concerns have been addressed! Of course, our paper will be revised accordingly to include experimental results in our rebuttal and to clarify confusing points. Thanks to the reviewer.

---

### Official Review · Reviewer_XQBy · 2023-07-04

**Soundness:** 3 good
**Presentation:** 3 good
**Contribution:** 2 fair
**Rating:** 5
**Confidence:** 3

**Summary:**

This paper presents an approach to enhancing the transferability of adversarial examples in a black-box scenario. Previous studies have typically followed a two-stage process involving the derivation of effective guiding directions and subsequently maximizing the perturbation magnitudes accordingly. Unfortunately, this approach often deviates from the intended guides, resulting in suboptimal performance. In contrast, the authors introduce a one-stage method that utilizes a decayed perturbation function and an analysis tool to investigate the sources of deviation. Through extensive experiments, the authors demonstrate the effectiveness of the proposed approach.

**Strengths:**

1. The paper is well-written, and the motivation is sound.
2. The analysis provides insights and could serve as a tool for future work.
3. Extensive experiments verify the effectiveness of the proposed method.

**Weaknesses:**

1. Despite the empirical effectiveness, the proposed method is not well-explained. Or, at least, it is unclear to me. The authors may like to elaborate more on how the "dual" equation (Eq. 3) is derived and provide insights into the equation. There seem to be some alternatives. For example, one can maximize $\mathcal{L}(h(x+\Delta x))$. What would be the role of $h(x)$ in this formulation?

2. Following the previous point, the proposed method looks incremental or empirical finding to me if it lacks proper insights.

3. In L152, what is the link to the mixup? The authors mentioned it, but it needed further discussion in depth.

4. In Section 4, the analysis lacks the baseline ILA. The audiences will be interested in whether the proposed method improves the prior work as motivated by the authors in the introduction.

**Questions:**

1. The experiment settings seem to be different from prior work. What does the *different* mean in Sec 5-1? Why didn't the authors report the same architecture attack performance?
2. Meanwhile, the performance of ILA++ reported in the paper is different from the original paper. What would be the root reason for it?

**Limitations:**

1. The authors do not report the variance of the experimental results.

---

> ### Author Rebuttal · Authors · 2023-08-09
>
> Thanks for your feedback. Our responses to the comments are provided as follows.
> &nbsp;
>
> > Despite the empirical effectiveness, the proposed method is not well-explained. Or, at least, it is unclear to me. The authors may like to elaborate more on how the "dual" equation (Eq. 3) is derived and provide insights into the equation. There seem to be some alternatives. For example, one can maximize $L (h(x+\Delta x))$. What would be the role of $h(x)$ in this formulation?
>
> **Response:** Our motivation is summarized as follows to make it clearer.
> * The common belief in intermediate-level attacks, _e.g._, ILA, ILA++, and NAA, is that a larger magnitude of intermediate-level perturbations along the directional guide leads to improved transferability of adversarial examples. These methods achieve their goals via a two-stage mechanism, _e.g._, performing I-FGSM first to obtain the directional guide and enlarge the projection of the intermediate-level perturbation onto the directional guide.
> * However, in this paper, we point out that the directional deviation of intermediate-level perturbation from the guide, even subtle, does great harm to the transferability. Therefore, we seek the intermediate-level perturbation $h(\mathbf{x}+\mathbf{\Delta x}) - h(\mathbf{x})$ that is in an adversarial direction $\mathbf{v}$ and to possess larger magnitudes compared to $\mathbf{v}$, simultaneously.
> * It is achieved by defining $\mathbf{v} =  \frac{1}{\gamma} (h(\mathbf{x}+\mathbf{\Delta x}) - h(\mathbf{x}))$ as the directional guide and optimizing it to be adversarial in Eq. (3) in the paper. From such a definition, we know that the magnitude of the achieved intermediate-level perturbation naturally shows a larger magnitude than $||\mathbf{v}||$, thus the goal is achieved.
>
> We appreciate the suggestion about alternative options, but $L(h(\mathbf{x}+\mathbf{\Delta x}))$ (if $L$ is defined to be the cross-entropy loss as in our paper) can NOT be directly optimized. Not sure if the reviewer actually meant to say $L (g(h(\mathbf{x}+\mathbf{\Delta x})), y)$. If yes, we would like to remind the reviewer that this is equivalent to the baseline attack, _i.e._, I-FGSM, and it is not beneficial to transferability. We agree that there may exist alternatives, yet, from our perspective, the chosen formulation is the most straightforward and effective implementation accordingly to our motivation.
> &nbsp;
>
> > In L152, what is the link to the mixup? The authors mentioned it, but it needed further discussion in depth.
>
> **Response:** As has been discussed in lines 151-153 and lines 172-178, after rewriting the objective, our solution resembles performing mixup on the intermediate-level feature representations between adversarial and benign examples. Yet, in our solution, there is no randomness for $\gamma$ and the mix ratio $1/\gamma$ of the optimized subject is suggested to be relatively small ($\leq 0.5$). If we follow the mixup to use a random mix ratio sampled from a Beta distribution, then the optimization of adversarial examples hardly converge, as the ratio on the optimized subject change drastically during optimization.
> &nbsp;
>
> > In Section 4, the analysis lacks the baseline ILA. The audiences will be interested in whether the proposed method improves the prior work as motivated by the authors in the introduction.
>
> **Response:** We would like to politely remind the reviewer that we have discussed ILA from the perspective of gradient alignment in Section 4 already (specifically, in lines 228 - 234). Moreover, we have conducted comprehensive empirical comparisons between our method and other intermediate-level attacks (including ILA++ and NAA) in Section 5. These methods outperform ILA according to our experimental results and similar results in many previous papers. To be concrete, in our experiment, ILA achieves an average attack success rate of 44.80% on ImageNet, while ILA++, NAA, and our solution achieve 46.99%, 46.36%, and 57.06%, respectively.
> &nbsp;
>
> > The experiment settings seem to be different from prior work. What does the different mean in Sec 5-1? Why didn't the authors report the same architecture attack performance?
>
> **Response:** In Section 5.1, since a VGG-19 is used as the substitute model, we adopt an independently trained VGG-19 which has different weights but the same architecture as one of the victim models to make the evaluation more practical. Note that, in a black-box setting, it is less likely that the victim adopts exactly the same model as the substitute model, but it is more possible that the architecture is the same.
> &nbsp;
>
> > Meanwhile, the performance of ILA++ reported in the paper is different from the original paper. What would be the root reason for it?
>
> **Response:** As has been discussed in lines 257-258, this is because of differences in pre-processing. In this paper, we don't crop images before feeding them to the models, following some recent work [1][2], while, in the ILA++ paper the images were cropped.
> &nbsp;
>
> > The authors do not report the variance of the experimental results.
>
> **Response:** For compared methods that involve randomness, the standard deviation of their multiple run results is relatively small (generally smaller than 0.30% to be specific) compared to their performance gap, thus it does not affect the conclusions. We will add such results to our paper.
> &nbsp;
> &nbsp;
>
> [1] Jiadong Lin, et al. Nesterov Accelerated Gradient and Scale Invariance for Adversarial Attacks. In ICLR 2019.
> [2] Xiaosen Wang, et al. Admix: Enhancing the Transferability of Adversarial Attacks. In ICCV 2021.

---

> > ### Comment · Reviewer_XQBy · 2023-08-12
> > **Response to the authors**
> >
> > I appreciate the detailed and informative response, which has clarified most of my concerns. I will raise my score after. The reason and further comments are listed below.
> >
> > > We appreciate the suggestion about alternative options ...
> >
> > I apologize for the typo. I was actually contemplating the substitute for the term $h(x+\Delta x) - h(x)$ in Equation (3) and meant to know whether it is the optimal choice. But, I acknowledge that it is minor and agree with the authors' motivation on magnitude maximization while retaining directional information.
> >
> > > Link to mixup
> >
> > After reading the response, I still found it a weak link, but it's also minor. I respect the authors' opinion to keep it.
> >
> > > In Section 4, the analysis lacks the baseline ILA. The audiences will be interested in whether the proposed method improves the prior work as motivated by the authors in the introduction.
> >
> > I would like to further elaborate on this point. It would be of interest to readers to witness a quantitative comparison between the proposed method and ILA concerning the alignment of gradient angles, which could be incorporated into, e.g., either Figure 3 or 4. As highlighted in Section 3-1 (especially around Line 100), the deterioration in ILA's performance stems from deviations in directional angles, while the proposed method addresses the issue via a novel objective. Despite the comprehensive experimental validation of attack performance, this particular aspect is somewhat unverified within the paper.
> >
> > Beyond these comments, the paper seems novel and provides extensive experiments proving its effectiveness.

---

> > > ### Author Response · Authors · 2023-08-12
> > > **Thanks to the reviewer**
> > >
> > > It is great to know that most of your concerns have been addressed! As for quantitative comparison between our ILPD and ILA, we appreciate the further elaborated suggestion and we will add more ILA results to our paper, in addition to what have been given in the rebuttal (i.e., ILA achieves an average attack success rate of 44.80% on ImageNet, while our solution achieve 57.06%).

---

> > > ### Author Response · Authors · 2023-08-17
> > >
> > > Dear Reviewer XQBy,
> > >
> > > Thanks again for your positive feedback to our rebuttal and for the intention of raising your score! However, it seems that currently the score (on the system) is still the same as pre rebuttal, and we would like to gently remind that the rating is now able to be altered. As there are three days left for author-reviewer discussion, we would be more than happy to address any remaining concerns if you do have.
> > >
> > > Best regards,
> > > Authors

---

### Official Review · Reviewer_pWZd · 2023-07-04

**Soundness:** 3 good
**Presentation:** 3 good
**Contribution:** 3 good
**Rating:** 7
**Confidence:** 4

**Summary:**

The authors first introduce the observation that traditional Intermediate-Level Attack (ILA) and some subsequent works deviate from the directional guides, resulting in sub-optimal transferability. With such observation, the authors propose Intermediate-Level Perutbration Decay (ILPD), a one-staged optimization that decays the intermediate perturbation strength in return for amplified directional guide, and thus higher adversarial transferability.
Experiments on ImageNet and CIFAR-10 show that the proposed ILPD method outperforms SOTA transfer-based attacks in different types of models as well as robust models.


**Strengths:**

1. Claims and speculations in the paper are well grounded by theoretical analysis as well as empirical studies. From the hypothesis of directional deviation in Figure 1, then the introduction of the ILPD and verification in Figure 2, studies and discussion are organized layer by layer, which is an enlightening experience to read through.
2. Apart from showing the analysis of ILPD, the authors extend the discussion to previous works such as ILA++, NAA and LinBP. I particularly like the authors’ attempt to replace $D_i$ and $\nabla_{z_g}L(\mathbf{z}_g, y)$ to study the transferability in the gradient aspect.
3. Although the modification the authors propose is a simple math trick by scaling up the influence from the directional guide, the optimization falls back to one-staged and no longer resembles lLA, which is sufficiently novel.
4. The experiments are conducted in a large variety of models, both normal and robust, both CNNs and ViTs. Ablation study and hyper-parameters study are performed to understand the method in an in-depth manner.


**Weaknesses:**

I could not spot any major weaknesses in this paper. The following points are mostly minor and understandable:

1. Although the authors perform evaluations on robust models, the choice of models does not align with what used to be evaluated in the previous works (i.e. the NeurIPS 2017 Competition, some models with ensemble adversarial training, etc.).

2. The number of attack iterations is fixed to be 100, which is quite large considering most baselines perform only 10-20 iterations in their experiments. Plus the observation in Figure 4(b), does it imply the proposed method has a slow convergence and requires more iterations to yield transferable attacks?

3. The performance of ILPD seems to be quite sensitive to the choice of $\gamma$. For example, Figure 2 suggests setting $1/\gamma \approx 0.5$ might probably obtain good transferability. However, in Figure 6(a), setting $1/\gamma = 0.5$ can have a success rate >15% lower than the optimal choice. For a new dataset, it is hard to determine a good starting $\gamma$ due to the difference in Figure 6(a) and 6(b).

#### Grammar mistakes:
- Line 148: the method **seems decays**
- Line 212: **a** I-FGSM example
- Line 214:  the difference **become**
- Line 254 **a** Inception v3

Here are only a few. The authors are recommended to check the paper thoroughly for the remainings.

---

In summary, this paper is very well-written and provides a lot of evidence to support its claims. The only weaknesses that I could identify are largely outweighed by its strengths. Therefore I recommend accepting this paper.


**Questions:**

1. Regarding the ILA++ baseline, I would like more clarification on the base attack in ILA++ for directional guidance. Since two attacks are applied in series, how are they balanced to sum up to 100 iterations?

**Limitations:**

Weakness #3 is briefly discussed in the paper. To me, the search for hyper-parameters will be the major limitation of this work.

---

> ### Author Rebuttal · Authors · 2023-08-09
>
> Thanks for your positive feedback. Our responses to the comments are provided as follows.
> &nbsp;
>
> > Although the authors perform evaluations on robust models, the choice of models does not align with what used to be evaluated in the previous works (i.e. the NeurIPS 2017 Competition, some models with ensemble adversarial training, etc.).
>
> **Response:** We appreciate the comment. The NeurIPS 2017 Competition provides a robust Inception v3 and a robust Inception ResNet v2 which is trained to be robust to an ensemble of adversarial examples. The performance on the robust Inception v3 has been shown in our paper already (in Table 3). We now provide comparative results on the robust Inception ResNet v2 (with ensemble adversarial training) as follows. It can be seen that the advantage of our method is still obvious.
>
> |  ensemble adv. train.   | I-FGSM | DI$^2$-FGSM | NAA | ILA++ | ILPD |
> |:---------------------:|:-------:|:---------------:|:----:|:------:|:-------:|
> | Incepction ResNet v2 |   6.28%  |   8.68%       | 8.06%  | 10.02% | **13.12%**   |
> &nbsp;
>
> > The number of attack iterations is fixed to be 100, which is quite large considering most baselines perform only 10-20 iterations in their experiments. Plus the observation in Figure 4(b), does it imply the proposed method has a slow convergence and requires more iterations to yield transferable attacks?
>
> **Response:** Despite some previous work used 10-20 iterations to evaluate transfer-based attacks, in practice, many attacks cannot converge well within only 10-20 iterations (as observed in many other papers, _e.g._, [1]). Thus, to ensure a fair comparison, we used 100 iterations with a step size of $1/255$ for each method to guarantee convergence of all methods, just like in many papers, _e.g._, [2][3].
>
> In Figure 3 and Figure 4, we used a smaller step size of $\epsilon / 100$ to observe the trend of overfitting before exhausting the perturbation budget as mentioned in the paper (lines 262-263). Given such a small step size, more iterations are required to demonstrate superiority indeed. However, in practice, our method does not exhibit slow convergence, and we have compared it with I-FGSM, NAA, and ILA++ on ImageNet as follows. We show how the average success rate of different methods varies with the maximum number of optimization iterations. The victim models are the same as those in Table 1 in our paper.
>
>   |          | 10 iterations    | 20 iterations    | 40 iterations    | 60 iterations    | 80 iterations    | 100 iterations   |
>   |:--------:|:--------:|:--------:|:--------:|:--------:|:--------:|:--------:|
>   | I-FGSM | 16.95% | 20.86% | 22.38% | 22.34% | 22.40% | 22.36% |
>   | NAA    | 35.11% | 42.78% | 45.27% | 45.91% | 46.16% | 46.36% |
>   | ILA++  | 41.76% | 45.98% | 46.54% | 46.88% | 46.94% | 46.99% |
>   | ILPD   | **42.16%** | **50.64%** | **55.50%** | **56.51%** | **56.83%** | **57.06%** |
> &nbsp;
>
> > The performance of ILPD seems to be quite sensitive to the choice of $\gamma$. For example, Figure 2 suggests setting $1 / \gamma \approx 0.5$ might probably obtain good transferability. However, in Figure 6(a), setting $1/\gamma=0.5$ can have a success rate >15% lower than the optimal choice. For a new dataset, it is hard to determine a good starting $\gamma$ due to the difference in Figure 6(a) and 6(b).
>
> **Response:** In the paper, we suggest using $\gamma \geq 2$ (_i.e._, $1/\gamma \leq 0.5$ line 172) to narrow the search for such a hyper-parameter. Figure 6 further shows that our method outperforms all its competitors with $1/\gamma$ in a wide range of $[0, 0.3]$ and $[0.1, 0.6]$ on ImageNet and CIFAR-10, respectively, showing that the choice of $1/\gamma$ is not a big concern to ensure its experimental superiority over all state-of-the-arts. In practice, the attacker could tune the hyper-parameters using a variant of the substitute model as the victim on a small validation set of, for instance, 200 examples, on their data to ensure performance. As for Figure 2, it adopts the same victim model as in Figure 1, which shares the same $h$ with the substitute model, thus might be slightly different from practical victim models and requires a different value of $1/\gamma$ to reach optimal attack. In general, the evaluated victim models in, for instance, Table 2 share a similar optimal $1/\gamma$ to be compromised fully.
> &nbsp;
>
> > Grammar mistakes.
>
> **Response:** Thanks for pointing out the typos. We will fix them in the updated version.
> &nbsp;
>
> > Regarding the ILA++ baseline, I would like more clarification on the base attack in ILA++ for directional guidance. Since two attacks are applied in series, how are they balanced to sum up to 100 iterations?
>
> **Response:** For previous intermediate-level attacks that require two-stage optimization, we do not count the number of iterations in their first-stage optimization in the 100 iterations. That is, if we count them all together, then these methods require more than 100 iterations to reach their performance reported in our paper.
> &nbsp;
> &nbsp;
>
> [1] Zhengyu Zhao, et al. Towards Good Practices in Evaluating Transfer Adversarial Attacks. In arXiv 2022.
> [2] Yi Huang, et al. Transferable Adversarial Attack Based on Integrated Gradients. In ICLR 2021.
> [3] Yiwen Guo, et al. Backpropagating Linearly Improves Transferability of Adversarial Examples. In NeurIPS 2020.

---

> > ### Comment · Reviewer_pWZd · 2023-08-10
> > **Thank you for the response**
> >
> > I would like to thank the authors for the responses and the extra experiments. My concerns regarding convergence rate and choice of $\gamma$ are well addressed.
> >
> > Just a clarification here: I believe the robust Inception v3 (from timm) used by the authors originally is trained by standard adversarial training (AT), which uses white-box attacks. By NeurIPS 2017 Competition, I indeed referred to the models
> > - Inc-v3$_{ens3}$
> > - Inc-v3$_{ens4}$
> > - IncRes-v2$_{ens}$ (added in the rebuttal)
> >
> > as used in citations [11, 12, 28, 42, 46] in the paper. These models are trained using **ensemble** adversarial training (EAT), which is claimed to be more robust against black-box attacks. Nevertheless, I don’t think the results of said models will influence the conclusion given the extensive experiments in other sections and the appended result of IncRes-v2$_{ens}$.
> >
> > Therefore, having no further concerns, I am inclined to keep my rating.

---

> > > ### Author Response · Authors · 2023-08-11
> > > **Thanks to the reviewer**
> > >
> > > We would like to thank the reviewer for responding to our rebuttal. It is great to know that your concerns have been addressed.

---

### Official Review · Reviewer_ncUa · 2023-07-09

**Soundness:** 3 good
**Presentation:** 2 fair
**Contribution:** 3 good
**Rating:** 6
**Confidence:** 3

**Summary:**

This paper notes that ILA (intermediate level attack), which maximizes projection onto a guide attack, suffers when the direction of the attack ends up differing from the direction of the guide, even when the projection is large. Thus, they propose that instead of allowing the direction to vary from a guide, we should simply look for attacks that have large intermediate magnitudes.

This motivates their approach (called Intermediate Level Perturbation Decayed). They then demonstrate that this approach results in significantly improved performance across a wide range of models.

**Strengths:**

Overall, I think the issue the authors identify with ILA is well motivated, and the improvements are fairly significant. In particular, the experiments investigating gradient alignment are fairly convincing about what ILPD is doing, as well as the analysis of existing attacks.

**Weaknesses:**

My primary issue with this paper was that I found the presentation confusing. Although the motivation for why we might prefer an approach that doesn't differ in direction from an original "guide" attack is solid, the actual motivation of the approach used was much less clear to me, and not particularly intuitively presented. From my understanding, the attack is essentially "simulating" reducing the strength of the perturbation at some intermediate stage, but this interpretation isn't presented in the paper.

In particular, upon first reading, this line was particularly confusing to me.

> seeks intermediate-level perturbations to be in an adversarial direction and to possess larger norms (or say larger magnitudes) compared to a directional guide in the same direction simultaneously

Since if there was a perturbation of the input that could lead to a strictly larger perturbation along the same direction as a guide, I would expect that to be already found by standard optimization procedures. I believe the part I missed from my first reading is that there is no "directional guide" - in comparison with ILA (a 2 step procedure), this is a 1 step attack.

Although I think I now have some intuitive understanding of the paper's attack, it doesn't quite match the author's explanation, and I think the paper would benefit from improving the clarity of section 3.2



**Questions:**

See above.

---

> ### Author Rebuttal · Authors · 2023-08-09
>
> Thanks for your positive feedback. Our response to comments about our presentation is provided as follows.
>
> **Response:** The common belief in intermediate-level attacks, _e.g._, ILA, ILA++, and NAA, is that a larger magnitude of intermediate-level perturbations along the guide leads to improved transferability of adversarial examples. Thus, these methods achieve their goals via a two-stage mechanism, _e.g._, performing I-FGSM first to obtain the directional guide and maximize the scalar projection of the intermediate-level perturbation onto the directional guide.
>
> However, in this paper, we point out that the learning objective of these methods inevitably leads to intermediate-level perturbations deviating from the guide, and such a directional deviation of intermediate-level perturbation, even subtle, does great harm to the transferability. Therefore, we seek the intermediate-level perturbation $h(\mathbf{x}+\mathbf{\Delta x}) - h(\mathbf{x})$ whose direction is already aligned with an adversarial directional guide $\mathbf{v}$ and, simultaneously, possess larger magnitudes than it. This is achieved by defining $\mathbf{v} = \frac{1}{\gamma} (h(\mathbf{x}+\mathbf{\Delta x}) - h(\mathbf{x}))$ as the directional guide and optimizing Eq. (3) in the paper. In the paragraph below Eq. (3), we have mentioned that the method seems to reduce the strength of the intermediate-level perturbation during optimization, from a dual perspective, and this is actually why we call it intermediate-level perturbation decay (ILPD). The effectiveness of such an intermediate-level perturbation decay, from another perspective, is explained in Section 4.2.
>
> Although the method can be interpreted superficially as decaying intermediate-level perturbations during optimization, our in-depth motivation is to encourage the finally achieved intermediate-level perturbation to be larger than a directional guide and have the same direction as the directional guide.
>
> We are more than glad to follow the suggestions from the reviewer and revise Section 3.2 to make it clearer.

---

### Decision · Program_Chairs · 2023-09-21

**Decision:**

Accept (poster)

**Comment:**

The paper proposes a one-stage optimization method that improves the alignment between the directional guide and feature perturbations in crafting the black-box adversarial attack. The paper initially received somewhat mixed reviews of two Weak Accepts, one Accept, one Borderline Reject, and one Reject. The primary concerns raised by the reviewers were about the clarity of the paper, the choice of the baseline methods and experiment settings, and comparisons with more surrogate models. The authors adequately addressed most concerns in the rebuttal, hence all reviewers recommend acceptance. The AC agrees with the reviewers’ decision. The authors should update the camera-ready paper to include the changes suggested by the reviewers during the review process.